# Leader emergence and affective empathy: A dynamic test of the dual-hormone hypothesis

**John G. Vongas**[1]☯*, **Raghid Al Hajj**[2]☯, **John Fiset**[3]

**1** Department of Management, Ithaca College School of Business, Ithaca, New York, United States of America, **2** Department of Management, John Molson School of Business, Concordia University, Montreal, Quebec, Canada, **3** Department of Management, Sobey School of Business, Saint Mary's University, Halifax, Nova Scotia, Canada

☯ These authors contributed equally to this work.
* jvongas@ithaca.edu

**Data Availability Statement:** The data associated with this study is publicly available at https://osf.io/xprm2/.

**Funding:** We acknowledge support by the Fonds Québécois de la Recherche sur la Société et la

## Abstract

Personal distress is a building block of empathy, yet has received scant attention in studies of individual differences in leadership. We investigate whether the effect of leader emergence on men's distress is influenced by their personalized power motive (p Power) and changes in their testosterone (T) and cortisol (C) levels. In an experiment involving 96 males, p Power modulated the direction and intensity of T change in emergent leaders, with high p-Power leaders showing a more positive T change compared to their low p-Power counterparts. We also conducted a dynamic test of the dual-hormone hypothesis in which participants' changes in T and C interacted to produce differences in personal distress. Contrary to expectations, positive changes in T were associated with increased distress at negative changes in C. Given that high T and low C are associated with leadership, we explain these findings and question the assumption that personal distress represents a shortcoming in leaders.

## Introduction

Leader emergence is the extent to which someone is viewed as a leader by others who possess limited knowledge of that individual's performance [1], and it results either from one's ascribed physical and dispositional traits or through achievements that signal competence [2]. Among the myriad of prototypical qualities that correlate with emerging leaders, two stand out as antithetical to one another: dominance and empathy [3]. Whereas dominance is an individual's egoistic desire to pursue and fulfill one's agenda, often at the expense of others, empathy is a concern for others and their wellbeing [4, 5].

Scholars have argued that prospective male leaders are expected to be dominant (or agentic) and female leaders to be empathic (or communal) because cultural norms shape the behavior of each sex into a set of learned social roles [6]. This conceptualization, however, is limiting for several reasons. First, it neglects to take into account the complex nature of traits, states, and behaviors that any given individual expresses on a daily basis [7]. In other words, a person's repertoire of traits or behaviors could encompass both agentic and communal characteristics simultaneously. For example, a dominant leader who wishes to exercise control over rivals in a

Culture (dossier # 135286). The funder had no role in study design, data collection and analysis, decision to publish, or preparation of the manuscript.

**Competing interests:** The authors have declared that no competing interests exist.

competitive field may also express nurturance when a group member faces a dire predicament. Second, it focuses on characteristics or traits as determinants of leadership emergence and neglects to question the degree to which newly appointed leaders are empathic to others. Finally, it is guided by the conventional wisdom that socialization is responsible for producing the observable individual differences in cognitive, affective, and behavioral outcomes between emergent leaders and nonleaders without considering evolutionary biological accounts of leadership [8].

To address these limitations, we apply evolutionary theory to investigate the extent to which individuals who emerge as leaders after an achievement-based competition differ from nonleaders in a core facet of empathy, namely personal distress. Thus, we veer away from studying individual differences as antecedents of leader emergence and instead explore an empathic outcome manifested *after* a leadership role is assumed. Personal distress is the intensity with which a person experiences anxiety upon witnessing another's pain [9], and the importance of this outcome is underscored by the fact that organizations are often sites of interpersonal pain and suffering [10]. Some organizational scholars have argued that feeling empathy for the pain of others is one way in which leaders help followers overcome aversive events and frame them into opportunities that inspire action [11].

In pursuing this goal, we explore two classes of individual differences expected to influence personal distress, one endocrinological (i.e., testosterone and cortisol) and the other psychological (i.e., implicit power motivation). Research involving either testosterone (T), cortisol (C), or their interplay has produced inconclusive findings on the typical hormonal profile of leaders [12–15]. One reason for this inconsistency lies in the fact that these studies focused on basal or resting T and C levels as opposed to measuring hormonal dynamics, which match the ebb and flow of human sociality. A recent empirical evaluation of the dual-hormone hypothesis by Grebe et al. [16] is an example of how researchers, even in the presense of longitudinal hormonal data, gravitate towards the use of basal values in testing the hypothesis's predictions. The authors concluded that the robustness of such predicitons is questionable but do not extend their analysis to dynamic hormonal changes. Therefore, we add to the literature by quantifying both T- and C-level changes among newly emergent leaders and nonleaders as these hormones affect empathic responding [17–19]. Consistent with a recent review which suggests that psychological consequences emanating from hormonal *changes* in response to status competitions are yet to be fully known [20], we also examine individuals' implicit power motive, i.e., their nonconscious desire to influence others and society at large, which is associated with both hormones as well as the motivation to lead [21–23].

Finally, we confine our work to men because distinct selective pressures are thought to have contributed to sex differences in leader emergence [24, 25] and empathy [26, 27]. Sex differences in competition have also been reported, particularly in the workplace, such that males and females appear to prefer competing intrasexually, i.e., with members of their same-sex cohort [28, 29]. We also elected to study males because sexually dimorphic endocrine systems result in lower and less variable T levels in females [30, 31] and, more importantly, because T appears to function differently in each sex; whereas T plays a role in both priming and learning reinforcement in males, it plays solely a priming one in females which may explain why they tend to persist more than males following a failure [32, 33]. Finally, we chose to focus exclusively on males because sex differences have been observed in C secretion levels among individuals who anticipate challenges [34].

## Personal distress: An affective component of empathy

Ever since Dymond's [35] early attempt to define empathy in modern psychological science, researchers have conceptualized it either as a cognitive ability (e.g., mentalizing or perspective

taking), as an affective entity (e.g., personal distress, emotional contagion) or as a prosocial behavior (e.g., concern, compassion) [36]. Recent research in neuroscience has provided evidence in the form of brain images to demonstrate that empathy involves socio-cognitive and socio-affective processes [37, 38], leaving the behavioral processes to other disciplines. Today, most scholars–irrespective of field–concur that empathy is multifaceted comprising all three orientations (i.e., thinking about others, sharing others' feelings, and caring about others; see [39]). Thus, researchers who study empathy may choose to focus on any one. A key dimension of empathy, personal distress is an aversive reaction whenever one perceives cues related to another's grief or anxiety [40]. More specifically, Davis defined it as a "self-oriented response characterized by feelings of anxiety and unease to distressed targets" [41, p. 106]. According to some, personal distress signals solidarity because, when group members share similar emotions, they indicate to one another that they possess a mutual understanding and that they have the same interests [42]. Given the complexity of feigning one's emotions relative to the use of words, personal distress can therefore be interpreted as a sincere way of gauging individuals' affective synchronicity with one another (i.e., being emotionally 'in sync').

One problem lies in that personal distress has been likened conceptually with sympathy and compassion, making it difficult to achieve consistent relationships between empathy, broadly defined, and other variables of interest. Sympathy is an emotional response based on an understanding of another's emotional state and involves feeling the same emotion that another person feels. Thus, sympathizing with another person means "feeling distressed *as* the other" [43, p. 8]. In contrast, personal distress triggered from witnessing another's anguish is focused more on the self rather than on others [44] and involves "feeling distressed *by* the other" [43, p. 8]. Observers who are experiencing personal distress are said to be more worried about lessening their own vicarious emotional arousal than with reducing the target's distress [40]. Compassion departs from this affective sharing seen in both sympathy and personal distress, and instead is characterized by caring that motivates one to assuage the suffering of another [45]. Thus, compassion is akin to "feeling distressed *for* the other" [43, p. 8] and calls for improving the other's wellbeing while keeping an emotional distance.

Compared to other facets of empathy like compassion [46], emotion recognition [47], and perspective taking [48], less is known about how personal distress relates to leadership. Some scholars are supportive of leaders who openly reveal their distress, claiming that affective displays increase a leader's trustworthiness among followers [49]. Others are more critical, proffering that demonstrations of distress contribute to perceptions of leader self-focus thereby signaling a lack of interest in followers [50]. Whether beneficial or detrimental to leaders and/ or their followers, personal distress seems to be affected by one's biological heritage [36, 41], which raises a key question: What function does personal distress serve in leader-follower relationships?

## Evolution, status attainment, and leader empathy

Throughout humankind's evolution, the coordination of group activities and dissemination of expertise required to achieve collective goals stood as adaptive problems that led to the development of specific leader-follower psychological processes [8]. Empathic responses are a set of such processes that evolved within many interpersonal domains, including that of leadership and followership. For example, adopting other people's perspectives and feeling sympathy at their misfortune aided leaders in gauging the emotional pulse of a group when motivating members toward a common goal. The evolutionary mechanism explaining how these empathic processes emerged resides in Trivers's [51] reciprocal altruism theory, which purports that selection acted on the recurring interactions between members who forged

exchange relationships to facilitate each other's biological fitness. Should the fitness benefits that one party received be larger than the costs the other sustained in providing aid, then those who engaged in this reciprocation outreproduced those who did not, causing this type of helping to proliferate in a population. Thus, psychological mechanisms for assisting nonkin evolved around the principle of mutual reciprocity over time. Altruism would not have evolved into the current form without cooperating members' capacity to empathize with one another, and studies have shown that eliciting participants' empathy leads to more dyadic cooperation [52]. However, many qualitative accounts hint that transitioning to a high-status position, such as a leadership role, brings with it an ostensible reduction in empathy [53].

The idea that achieving social status undermines one's empathy is not new. In nonhumans, primatologists have noted that subordinates shift their attention uniformly toward the group's high-ranking member, suggesting that other-oriented concern tends to be associated more with followers than leaders [54]. In humans, psychologists and organizational scholars have observed that executives often resort to self-adulation and the exploitation of others [55]. Overconfidence appears to be a predictor of leader emergence as evidenced by the babble hypothesis [56], whereby one who interjects the most during a group interaction is most likely to emerge as leader. Overconfident leaders who seize opportunities and make assertive decisions enjoy the deference of followers [8], which implies that they are somewhat permitted to ignore individual concerns. In recent years, Van Vugt and colleagues advanced the male-warrior hypothesis which holds that selective pressures tailored men's proclivity to partake in intergroup conflict and, in particular, favored males over females as chosen coalitional leaders against rival groups [57]. One implication of this hypothesis, albeit subtle, is that 'capable' (i.e., aggressive) warriors who were successful at safeguarding or exploiting valued resources would likely not have been characterized as such if they were excessively empathic toward outgroup members. This skepticism around the exaggerated expression of empathy may have prompted Darwin to suggest that "He who was ready to sacrifice his life [. . .] rather than betray his comrades, would often leave no offspring to inherit his noble nature" [58, p. 157].

Social status is not only the extent to which one possesses influence in controlling coveted resources [59], but also the respect and admiration that one receives by others [60]. Hence, two strategies appear to be involved in attaining status in newly formed groups. The first, dominance, is the ability to inflict harm on others as a means of securing rank. Dominant leaders, therefore, tend to use coercion and intimidation when dealing with followers and, as a result, induce fear. The second strategy, prestige, is the ability to confer benefits on others through competence. By sharing knowledge and skills, prestigious leaders receive follower praise and esteem [61–63]. There is now a consensus that these two routes may operate concurrently in leadership contexts [61, 64]. That is, dominance and prestige are often both reflected in competency-based competitions in which individuals routinely participate to gain status and demarcate themselves as leaders [65]. Those who emerge as leaders based on prestige, rather than dominance, are more likely to prioritize the wellbeing of the group and maintain strong relationships with group members as these serve to undergird the status of the leader and encourage the development of group norms that foster prosocial motivation [63]. For nearly 40 years, researchers have found that changes in men's social status emanating from competitions are correlated with changes in their T levels [66].

## Leader emergence, testosterone, and implicit power motivation

In group-based societies, leadership is the influence with which one person induces others to think and act in particular ways [67]. It is also a transaction between a person of authority and the surrounding social setting [68]. Leadership and influence are thus interwoven concepts

because power and status differences distinguish some members from others in the hierarchy. Whereas power enables access to resources [69], status is the level of positional prominence within a group [42]. In organizations, candidates contend for leadership roles and their eventual ability to exercise power is often dependent on accrued status.

The evidence that men's T is sensitive to changes in hierarchical rank is consistent with the notion that T evolved out of a struggle for dominance required to solve the adaptive problem of survival through resource acquisition [70]. The relationship between status indicators like leadership and T change, however, does not operate in a linear fashion as prevailing theories suggest [15]. One such theory is Mazur's [71] biosocial model, which holds that status elevations and relegations elicit T surges and reductions, respectively. It also posits a reciprocal relationship in which elevated T facilitates further status-seeking behaviors while reduced T impedes such behaviors. Although some studies support these predictions, others yield contradictory findings [66].

One explanation for this inconsistency lies with an important individual difference which modulates the relationship between male status attainment and T change, namely implicit power motivation [18, 72]. In contrast to social power which derives from one's hierarchical position, the power motive operates outside one's consciousness and reflects one's desire to influence others not only through control and coercion, but also via public praise and recognition [73]. Individuals scoring high in implicit power see the prospect of impacting others as hedonic whereas those scoring low do not, a distinction that explains why the former are perceived to be more influential and more likely to seek leadership opportunities [74].

McClelland [75] differentiated implicit power into two major subcategories, one of which is personalized power or p Power. This form represents one's egoistic need to sway others for self-aggrandizing purposes and often surfaces during competitions. As he put it, those high in p Power are "turned toward seeking to win out over active adversaries" [75, p. 36]. Thus, p Power is critical in contests for power and status–such as when leadership positions are at stake–and would likely underlie the leader emergence-T link. Hormonal studies have supported this conjecture. Wirth et al. [23] found that high p-Power individuals who lost status exhibited a rise in C, as did their low p-Power cohort who gained status, indicating that achieving and failing to achieve dominance is stressful for low and high p-Power individuals, respectively. Some avow that the reticence people feel about taking on status may stem from the absence of a dominant personality. For example, Josephs and colleagues [76] used basal T as a correlate for an individual's status fixation. When given roles of varying status, 'submissive' (low-T) and 'dominant' (high-T) participants showed similar patterns of performance and functioning. In other words, those whose T level was mismatched with their corresponding status had the worst performance and reported the most negative affect. Finally, other research has shown a more direct role that p Power plays in how status-based competitions contribute to male T changes [22, 72]. In these studies, high p-Power men had T increases following status victories and T decreases following defeats, while low p-Power men showed post-victory T reductions.

Many studies involving physiological outcomes and competitive rivalries, however, were not designed to address leadership contexts per se. Moreover, of the studies investigating empathy in leaders, the vast majority have focused on either its cognitive or behavioral subdimensions leaving affective empathy largely unexplored [77]. Given that this emotional component has received criticism for interfering with rational decision-making [36], it makes it all the more important to scrutinize it among emergent leaders.

## Leader emergence, hormones, and personal distress

Some insights about the relationship between T reactivity and personal distress can be observed in the wider psychological literature. Male-male competition in survival and

reproduction contributed to sex differences in the intensity with which emotions are experienced [78]. For instance, repressing fear and anxiety is adaptive to some degree because publicizing these emotions would have diminished a competitor's status within a group [79]. Research has shown that T reduces people's affective responses [80], and explanations for this phenomeon range from T facilitating one's composure during conflict to priming one for confrontation. When studying the relationship between status-induced T changes and personal distress, however, it is important to consider one key contextual feature of status rivalries: psychological stress.

Cortisol (C) is a glucocorticoid hormone produced by the adrenal cortex in response to stress. In general, elevated C is associated with social avoidance and anxiety, whereas low C is related to approach strategies and reduced stress. Stressful contexts, like status pursuits, also bring about changes in T [81]. Under stress, male T changes result from two opposed effects which act on T release, one coming from the adrenal glands and the other from the sympathetic nervous system (SNS). In the former, stress stimulates the hypothalamic-pituitary-adrenal axis to signal to the adrenals to release C which, in men, inhibits T production [81, 82]. In the latter, the SNS releases epinephrine which regulates the fight-or-flight response and stimulates T secretion. When witnessing others' distress, high-epinephrine (E) individuals are more likely to insulate themselves from the distress compared to those low in E [83]. Since a status victory helps to promote E secretion and aids in T's release, it is likely that individuals being promoted to a leadership role as emergent leaders will prefer to seek rewarding stimuli and avoid threatening ones.

Until recently, studies reporting the association between T and emotions or behavior failed to acknowledge T × C interaction effects. To address this, the dual-hormone hypothesis [13] has been proposed to account for inconsistent findings involving T and behavior. It holds that T's influence on dominance-seeking behavior depends on circulating C levels. In particular, it specifies that T will be positively related to dominance behavior only when C is low; when C is high, T's effect on dominance behavior should be inhibited. Many studies have supported this hypothesis [20, 84]. One such study had graduate business students complete a self-assessment of trait empathy and findings showed that empathic impairment was associated with a hormonal signature comprised of high T and low C [19]. The dual-hormone hypothesis also claims that high T and low C are indicative of one's social power. Mehta and Josephs [13] videorecorded undergraduates following random assignment to either a leader or follower role-play scenario, after which assistants rated them on dominance (e.g., assertiveness). They found that participants high in T and low in C demonstrated the most dominant leadership behavior.

## The present investigation

The question remains as to how hormonal fluctuations that emanate from a leadership contest are related to the personal distress of individuals who differ in implicit power motivation. Therefore, the focus here is to explore psychophysiological phenomena (p Power, T, and C) that underlie the association between leader emergence and a critical affective component of empathy, namely personal distress. The power motive has been linked with behaviors that enable individuals to seek and maintain status, and leadership represents a relevant context for its study [21, 85]. The same motive has been involved in differentially predicting changes in male T levels following competition [22, 72]. Since competitions resulting in the selection of a leader can be thought of as social status contests, we expect that the effect of leader emergence on T change will be moderated by p Power. That is, emergent leaders having a high p Power will show a more positive T change than their low p-Power cohort.

Using a controlled laboratory experiment, we randomized male participants in two conditions stemming from a leadership contest manipulation (i.e., emergent leaders and nonleaders). Pre- and post-manipulation saliva samples were collected from which both T and C changes were assessed. Participants were then asked to report how they felt when witnessing others in various perturbing circumstances. Guided by the dual-hormone hypothesis [13], we expect that increases in T will be associated with reduced personal distress when levels of C change are low. Finally, we also argue that the influence of leader emergence on personal distress could be carried over through T's reactivity to changes in individuals' perceptions of themselves as leaders. Given the plethora of research on outcomes of leader emergence employing nonbiological mechanisms, we argue that the hormonal mechanism explored here is a missing piece in understanding how an empathic outcome like personal distress may become manifest.

## Methods

### Participants

Male participants were recruited from a large undergraduate cohort enrolled at Concordia Univerity's John Molson School of Business, located in Montreal, Canada, in exchange for course credit. All procedures, including recruitment, received ethics approval (file # 10000642-UH2012-080). Each participant was informed that the purpose of the research was to explore men's hormonal changes across various exercises played either alone or in groups of other people, and to understand how these social interactions affect their behaviors in the short term. After giving their informed consent, several precautions were taken to mitigate spurious hormonal fluctuations. First, participants were asked to abstain from tobacco and alcohol consumption prior to experimentation [86]. We also screened for endocrine dysfunction and anabolic steroid use. Third, again prior to experimental testing, we asked participants to dispose chewing gum or candy and to rinse their mouths with water to minimize test tube contamination. Finally, we asked them to store mobile phones to prevent verbal confrontations from influencing hormonal elicitation [87]. Since we could not locate previous research reporting effect sizes usable in generating sample size estimates for the testing of the dual-hormone hypothesis in the context of personal distress, we aimed for a sample size equivalent to that reported by studies in the field that included the most participants. As such, and basing our estimates on previous research [66], we aimed for approximately 100 participants. The initial sample included 102 undergratuate males ($M = 21.50$ years, $SD = 2.74$). We discarded three participants with missing data because their item nonresponse exceeded 5% [88]. We also tested for outliers using box-plots and removed two participants as baseline T levels were above three standard deviations from the mean and one had produced inadequate saliva to measure both C and T. Therefore, 96 men ($M = 21.58$ years, $SD = 2.79$) constituted the final sample for the current study. The recent meta-analysis by Geniole [66] reported over 60 effect sizes from 49 different manuscripts dealing with the effects of competitive rivalries on T concentrations, and revealed that our final sample size was greater than those in 96% of studies (with the exceptions of two, e.g., 105 males in [89]; 106 females in [90]). Based on recommendations for handling outliers in hormonal data [91], we ran all analyses using the full sample and one without outliers. Results were similar in significance and direction.

### Procedures and materials

Participants were asked to provide data on two separate days. On Day 1, they completed the Big Five [92] and the Dark Triad [93] personality questionnaires. Next, they were introduced to the Number Tracking Test (NTT), a timed cognitive task requiring a person to trace a

continuous line with a pen through consecutive ascending numbers on a grid filled with distractor numbers until an arbitrary number is reached [72]. Participants completed one NTT round and their score, timed in seconds, was used to pair others with similar scores during the leadership contest on Day 2. This is consistent with the recommended use of a matching variable that is theoretically related to the outcome variable [94], and ensures the methodological validity of the contest outcome manipulation given its rigging.

On Day 2, participants with similar scores on the trial NTT arrived at the laboratory between 9:55 am and 5:55 pm to control for diurnal hormonal variation due to circadian rhythms [95]. They sat at desks separated by a partition and were told that communication with one another was prohibited. Once seated, they rested for 10 minutes to quell any anxiety arising from ambiguity in the task and surroundings [96]. They then deposited their first saliva sample ($T_1$ = 10 min) into a 5-ml sterile plastic vial through passive drooling [97]. These vials were stored on ice and transferred to a freezer until the study was completed, at which time all samples were shipped using dry ice to a laboratory for assaying. Participants then engaged in the Picture Story Exercise (or PSE) meant to capture their p Power [98, 99].

After completion of the PSE, both participants were taken to another room where they were told they would be competing with another adversary on 12 rounds of the NTT. Several noteworthy modifications were made to the NTT's administration originally described by Schultheiss [72]. First, the dyads competed in the sole presence of a female moderator who oversaw the entire competition. Second, to ensure a successful leader emergence manipulation based on Judge and colleagues [1], participants were informed that the NTT represents a valid measure of leadership potential, future earnings, and likelihood of career success, and that it has been used by organizations to successfully predict future leaders. Third, they were required to sit with their backs to each other since face-to-face interactions by unacquainted males may elicit confrontational mechanisms that could affect T release [100]. Fourth, they were told that winning more NTT rounds than their rival would depend on their skill, thus rendering the competition to be 'up for grabs,' and preventing them from making other attributions (e.g., luck) which have also been shown to impact T [101]. Fifth, participants were informed that the winner (i.e., emergent leader) would be assigned to the role of group leader in a subsequent exercise and, conversely, those who did not secure a leadership role would act as subordinates to the emergent leader. Finally, once the competition was over, a brief ceremony was performed to declare and honor the winner after which participants were escorted to the original room. Once seated, the manipulation was again reinforced by providing the leader with an endorsed letter attesting to his successful performance and giving both competitors a mock organizational chart illustrating the winner or emergent leader in the top position and the loser or nonleader occupying a lower position along with other fictitious losing participants. These specific modifications to the NTT were made to render an enhanced competitive paradigm by featuring both a social-evaluative threat (i.e., one's ability can be negatively judged by others following a loss) and a challenging quality, two key characteristics of potent stressors [102].

Unbeknownst to participants, each received a different version of the NTT task; those assigned randomly to the winning or emergent leader condition received nine shorter (i.e., easier) sets of NTT puzzles from the total 12. This rigging ensured consistency of the competitive outcome. Upon completion, and after having been guided to their original room, they were asked to sit and provide another saliva sample ($T_2$ = 70 minutes) approximately 10 minutes after the leader was announced. Finally, they concluded the study with a paradigm meant to assess their personal distress as detailed in the following section.

Before exiting, additional suspicion and manipulation checks were implemented. Each participant was asked whether he had guessed the study's hypotheses or had doubts surrounding

the study, and was asked whether he felt like a leader. Consistent with previous research [12], we verified the manipulation's effectiveness after the study's completion to avoid spurious hormonal fluxes. The responses of 91 participants matched their respective conditions, while those of four did not and one participant was uncertain, thus proving a successful manipulation. Given the social quality of the competition and its potential side effects on participants, measures were taken to mitigate any adverse effects during debriefing. They were informed that their result on the NTT was not due to skill, or lack thereof, and that the letter and organizational chart served only to accentuate prestige in the experiment. In sum, we explained why these steps were essential to the research, and reminded them of their right to withdraw from the study without losing course credit.

## Measures

**Testosterone and cortisol.**  Participants provided two saliva samples for T and C measures. Baselines were taken at $T_1 = 10$ minutes and post-experimentally at $T_2 = 70$ minutes. This is consistent with studies specifying when hormonal concentration changes appear in saliva after a social interaction [22, 96, 102, 103]. Frozen samples were thawed to room temperature and centrifuged at 3,000 rpm for 15 minutes. Following DRG International's kit instructions (https://drg-international.com/), duplicate 100 μL saliva aliquots were assayed. Using a Biotek Synergy™ plate reader at 450 nm, optical densities were determined. The batch in which participants' saliva samples were included yielded intra- and inter-assay coefficients of variation of 8.23% and 13.11% for T, and 9.80% and 9.61% for C, respectively.

**Personalized power.**  Personalized power (p Power) was assessed through the Picture Story Exercise (PSE) [99] a well-validated thematic content analysis tool that captures motive imagery from written text. Six digital photographs were presented randomly to each participant on a computer using Inquisit© software. The photographs depicted a range of social situations and included the following six: 1) ship captain; 2) bicycle race; 3) hooligan attack; 4) women in laboratory; 5) boxer; and 6) woman and man arguing. Each participant was shown one digital photograph for about 15 seconds after which he was given five minutes to freely write about whatever came to mind. Therefore, the total time taken for the PSE was roughly 30 minutes. These photographs were selected on the basis of effective power motive elicitation while minimizing respondent fatigue [98]. Two raters, including the first author, independently coded participants' stories for inferences of p Power using a modified version of Winter's original scoring manual [104]. In recent years, several researchers have articulated this system's coding flexibility [105, 106]. We modified the coding system in order to parcel out the specific motive, p Power. In doing so, we coded text fragments conveying wishes, intentions, and concerns for power under five specific categories: 1) strong and forceful actions impacting others (e.g., insulting, hitting, shouting); 2) attempts to control others or regulate their behavior (e.g., checking up on others); 3) efforts to influence, persuade, and convince others (e.g., arguing with someone either for or against a wish or idea); 4) impression management (e.g., preoccupation with fame, prestige, and social status); and 5) the desire to evoke emotional responses in others (e.g., surprising or startling others, making them laugh or cry). A sixth category included in Winter's [104] scoring system involves unsolicited helping of others (e.g., offering help, advice, or support without being asked to do so). Because our goal was to measure only the self-oriented personalized power motive (or p Power), we left this sixth category out of the assessment because it expresses an altruistic form of the power motive, or what has been termed socialized or s Power [75, see also 107].

Before the coding of PSE stories took place, each coder reached an agreement level of .85 during PSE training that encompassed practice materials, calibration, and test sets. For this

study, the interrater reliability or agreement score between the coders for stories was.90. Any remaining discrepancies between coders were resolved through discussion. A total of 1152 stories were coded, i.e., 96 participants × 6 stories each × 2 coders. Participants' overall p Power scores ranged from 2 to 20 (*M* = 9.60, *SD* = 4.00) and stories ranged from 142 to 1112 words (*M* = 602.18, *SD* = 201.62).

**Personal distress.**   Personal distress was measured using the International Affective Picture System [108]. Thirty-three photographs of individuals in various situations (e.g., sickness) were evaluated using the Self-Assessment Manikin, a valid nonverbal method used to capture ratings of emotional pleasure and arousal. Five manikins were arranged in order from very happy, to neutral, to very sad, and participants indicated how each photograph made them feel by selecting one of nine circles evenly spaced below the manikins (*M* = 5.07, *SD* = .37).

## Controls

**Psychopathy.**   One trait associated with an inability to experience personal distress is psychopathy [109]. We used Jonason and Webster's [93] (α = .73) four-item subclinical measure which uses a five-point scale ranging from 1 (*not at all*) to 5 (*very much*; e.g., "I tend to be unconcerned with the morality of my actions").

**Anxiety.**   Anxiety hampers the efficiency of cognitive processing by reducing attentional control [110]. We thus controlled for state anxiety using a shortened version of the Profile of Mood States [111] (α = .79). This measure asks participants to rate how they felt on a five-point scale ranging from 1 (*strongly disagree*) to 5 (*strongly agree*; e.g., "I feel on edge").

**Neuroticism.**   Since personal distress may be influenced by neuroticism [112], we controlled for it with a shortened version of the International Personality Item Pool [92] (α = .75) which uses a five-point scale ranging from 1 (*strongly disagree*) to 5 (*strongly agree*; e.g., "I have frequent mood swings").

**Statistical approach.**   Using the PSE to measure p Power requires that the word count in each story be controlled for, since lengthy stories have a greater chance of including more power references than brief ones. As such, we regressed p Power scores on the number of words and saved the standardized residuals for use in the analyses [113]. We used the same method when calculating changes in T and C. It is not unusual for researchers to use raw difference scores when measuring hormonal change [79]. However, due to concerns surrounding such scores [114], the 'regressor variable method' was also used by regressing post-manipulation hormone levels onto baselines and using the standardized residuals for analysis [115]. These residuals represent the hormonal change not attributed to pre-manipulation hormonal values. All analyses were repeated using the more common raw difference scores, and the patterns were consistent between the two analyses in both direction and significance. We present the results with the original variables in the section that follows.

We employed conditional process analysis Model 21 [116, 117] which features one variable (p Power) moderating the relationship between the independent variable (leader emergence) and the mediator (T change), and a second variable (C change) moderating the link between the mediator and the outcome (personal distress). Our conceptual model is presented in Fig 1. This approach calculates all paths simultaneously and is superior to other regression-based approaches particularly when the analysis includes interaction terms. Finally, it uses bootstrapping and repeated sampling with replacement to create multiple datasets from the original one for analysis.

## Results

Means, standard deviations, and correlations are presented in Table 1.

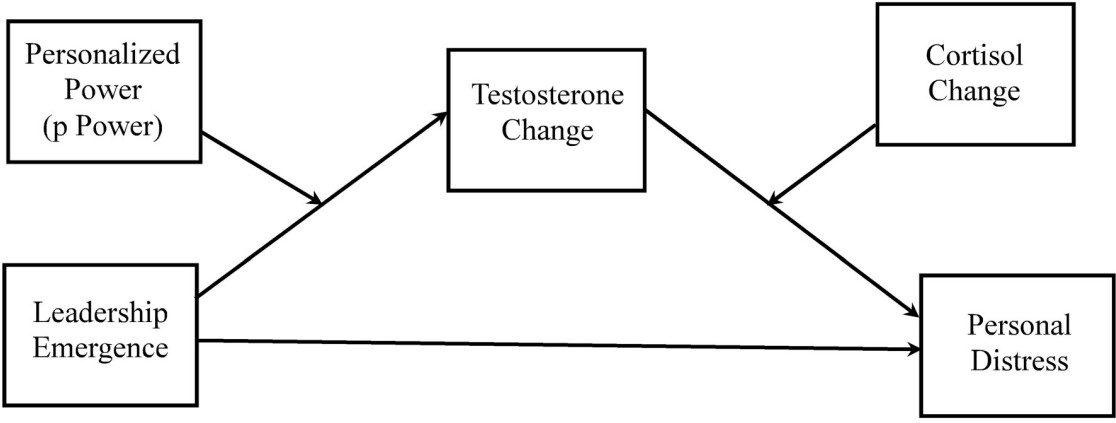

**Fig 1. Conceptual model.**

Correlations between pre- and post-manipulation C and T levels were moderate to high and significant ($r = .23$, $p = .03$ and $r = .89$, $p = .00$, respectively). Unexpectedly, the point-biserial correlation between leader emergence and p Power was significant ($r_{pb} = .25$, $p = .02$), with emergent leaders scoring significantly higher on p Power than nonleaders ($M_{\text{emergent leaders}} = .25$ and $M_{\text{nonleaders}} = -.25$, $t(94) = 2.48$, $p = .02$). This may have been due to either chance or an unforeseen bias in randomization. Sampling theory predicts a higher chance of significant pretest differences between conditions if the number of participants in each condition is low [94], as might have been the case here. To resolve this, we created another measure of p Power (p Power*) which is the standardized residual when regressing p Power on the manipulation condition. We repeated all analyses using this measure, representing the p Power score void of the effect of the emergence condition. No difference was found between the two

**Table 1. Descriptive statistics and study variable intercorrelations.**

| | Mean | SD | 1 | 2 | 3 | 4 | 5 | 6 | 7 | 8 | 9 | 10 | 11 |
|---|---|---|---|---|---|---|---|---|---|---|---|---|---|
| 1. Leader Emergence | .50 | .50 | | | | | | | | | | | |
| 2. Testosterone $T_1$ | 118.50 | 68.74 | -.05 | | | | | | | | | | |
| 3. Testosterone $T_2$ | 122.47 | 67.53 | -.02 | .89** | | | | | | | | | |
| 4. Cortisol $T_1$ | 2.98 | 1.98 | -.04 | .15 | .09 | | | | | | | | |
| 5. Cortisol $T_2$ | 2.81 | 1.82 | -.19† | .14 | .18† | .23* | | | | | | | |
| 6. Personal Distress | 5.07 | .37 | -.08 | -.18† | -.15 | -.12 | -.05 | | | | | | |
| 7. Personalized Power | 9.60 | 4.00 | .25* | -.09 | -.07 | .07 | -.25* | -.03 | | | | | |
| 8. Testosterone Change | 2.85 | 32.05 | .02 | .00 | .46** | -.09 | .14 | .05 | .01 | | | | |
| 9. Cortisol Change | -.20 | 2.36 | -.18† | .11 | .16 | .00 | .97** | -.03 | -.27** | .17† | | | |
| 10. Psychopathy | 1.99 | .69 | -.25* | .16 | .20† | .31** | .22* | -.17† | -.09 | .14 | .15 | | |
| 11. Neuroticism | 2.69 | .78 | -.18† | -.06 | -.03 | .03 | -.01 | .02 | -.09 | .08 | -.02 | .13 | |
| 12. Anxiety | 2.45 | .69 | -.18† | .05 | .06 | -.16 | .02 | -.07 | .02 | .07 | .06 | .09 | .36** |

*Note*: $N = 96$. In the correlation part of the table, testosterone change, cortisol change, and personalized power are standardized residuals and are thus unit-free. The means of the baselines, testosterone change, and cortisol change are in pg/ml (i.e., raw measures). Personalized power is also in raw scores.

†$p < .10$,
*$p < .05$,
**$p < .01$.

measures; thus, we report findings of the original p Power measure and present those of p Power* where needed. To test if this difference extends beyond p Power, we also tested whether participants differed on the Big Five and the other two controls. They were not significantly different for the Big Five and anxiety, but were so for psychopathy.

There was no significant difference between the groups in neuroticism ($M_{emergent\ leaders}$ = 2.55 and $M_{nonleaders}$ = 2.83, $t(94)$ = 1.79, $p$ = .08), extraversion ($M_{emergent\ leaders}$ = 3.53 and $M_{nonleaders}$ = 3.61, $t(94)$ = .53, $p$ = .60), agreeableness ($M_{emergent\ leaders}$ = 3.73 and $M_{nonleaders}$ = 3.72, $t(94)$ = .10, $p$ = .92), openness to experience ($M_{emergent\ leaders}$ = 3.28 and $M_{nonleaders}$ = 3.40, $t(94)$ = 1.12, $p$ = .27), conscientiousness ($M_{emergent\ leaders}$ = 3.32 and $M_{nonleaders}$ = 3.52, $t(94)$ = 1.42, $p$ = .16), and anxiety ($M_{emergent\ leaders}$ = 2.32 and $M_{nonleaders}$ = 2.57, $t(94)$ 1.80, $p$ = .08). The groups, however, showed a significant difference in psychopathy ($M_{emergent\ leaders}$ = 1.82 and $M_{nonleaders}$ = 2.16, $t(94)$ = 2.50, $p$ = .01).

## Conditional process analysis

When addressing our predictions, first, we found that p Power moderated the link between leader emergence and T-level change ($b$ = .64, $p$ = .00) (using p Power*, $b$ = .62, $p$ = .00) (Table 2).

To better portray the interaction we graphed the simple-slopes analyses for emergent leaders and nonleaders using a mean split on p Power. Fig 2 portrays the interaction.

Change in T for emergent leaders high in p Power ($M_{standardized\ residual}$ = .21 and $M_{raw\ difference}$ = 10.69 pg/ml corresponding to a 14.81% mean change in T, with change ranging from a 25.65% decrease to a 120.94% increase in T) was larger than that of emergent leaders with low p Power ($M_{standardized\ residual}$ = -.28 and $M_{raw\ difference}$ = -7.30 pg/ml corresponding to a -1.98% mean change in T, with change ranging from a 35.99% decrease to a 36.90% increase in T). Although in the expected direction, this change reached only marginal significance ($t(45)$ = 1.86, $p$ = .07). For nonleaders high in p Power, the change in T ($M_{standardized\ residual}$ = -.17 and $M_{raw\ difference}$ = -2.91 pg/ml corresponding to a 8.36% mean change in T, with change ranging from a 29.27% decrease to a 133.66% increase in T) was smaller than for those low in p Power ($M_{standardized\ residual}$ = .07 and $M_{raw\ difference}$ = 4.81 pg/ml corresponding to a 4.61% mean change in T, with change ranging from a 39.92% decrease to a 69.85% increase in T). Although it followed the expected direction, this change was nonsignificant ($t(46)$ = .83, $p$ = .41).

Our second prediction was guided by the dual-hormone hypothesis [13] and claimed that positive changes in T would be associated with reduced personal distress at negative changes in C. Although the interaction of C and T change was significant in predicting personal distress ($b$ = -.05, $p$ = .04) (Table 2), the direction of the interaction was opposite to that predicted. Once more, to better visualize the interaction, we graphed the simple-slopes analyses with a mean split on C change. The interaction at low and high levels of C change is seen in Fig 3. When the change in C is negative, a positive change in T was related to higher personal distress ($b$ = .13, $p$ = .05). However, at positive C change, it was associated with lower distress, although this relationship was found to be nonsignificant ($b$ = -.05, $p$ = .23).

Lastly, for our third prediction, to understand the extent to which T change is a mechanism linking leader emergence and personal distress, we employed PROCESS's 'index of moderated moderated mediation' (Model 21). This index tests mediation taking into consideration the effects of the two moderators, i.e., p Power and C change. A bootstrapped estimate of the confidence intervals around the index of moderated moderated mediation represents a test of whether the index is different from zero [116, 117]. The confidence interval for the index here did include zero (index = -.03, C.I. = [-.106,.005] (see Table 2) (using p Power*, index = -.03,

**Table 2. Conditional process analysis results.**

|  | Effect (SE) | *t* | *p* | 95% LLCI | 95% HLCI |
|---|---|---|---|---|---|
| **Outcome: Testosterone Change** |  |  |  |  |  |
| Leader Emergence | .16 (.22) | .74 | .46 | -.271 | .589 |
| Personalized Power | -.39 (.17) | -2.31 | .02 | -.731 | -.055 |
| Psychopathy | .23 (.15) | 1.53 | .13 | -.070 | .529 |
| Anxiety | .14 (.16) | .90 | .37 | -.175 | .461 |
| Neuroticism | .03 (.14) | .24 | .81 | -.244 | .312 |
| **Leader Emergence × Personalized Power** | .64 (.22) | 2.93 | .00 | .205 | 1.067 |
| **Outcome: Personal Distress** |  |  |  |  |  |
| Testosterone Change | .08 (.04) | 1.77 | .08 | -.009 | .160 |
| Leader Emergence | -.11 (.08) | -1.34 | .18 | -.269 | .052 |
| Cortisol Change | .02 (.04) | .45 | .65 | -.066 | .105 |
| Psychopathy | -.13 (.06) | -2.29 | .02 | -.247 | -.017 |
| Anxiety | -.03 (.06) | -.51 | .61 | -.150 | .088 |
| Neuroticism | .01 (.05) | .21 | .83 | -.094 | .116 |
| **Testosterone Change × Cortisol Change** | -.05 (.03) | -2.05 | .04 | -.106 | -.002 |
| **Index of Moderated Moderated Mediation (Using Process Model 21)** | -.03 (.03) | - | - | -.107 | .006 |

*Note*: N = 96. SE = standard error, LLCI = lower level confidence interval, HLCI = higher level confidence interval.

C.I. [-.101,.006]). As one could see, the interval's upper limit approaches zero, but did not include zero when the analysis was first conducted. Thus, caution is needed in result interpretation, and we err on the cautious side by noting that the prediction was not supported although preliminary evidence exists for such a mediation. Closer inspection of the indirect

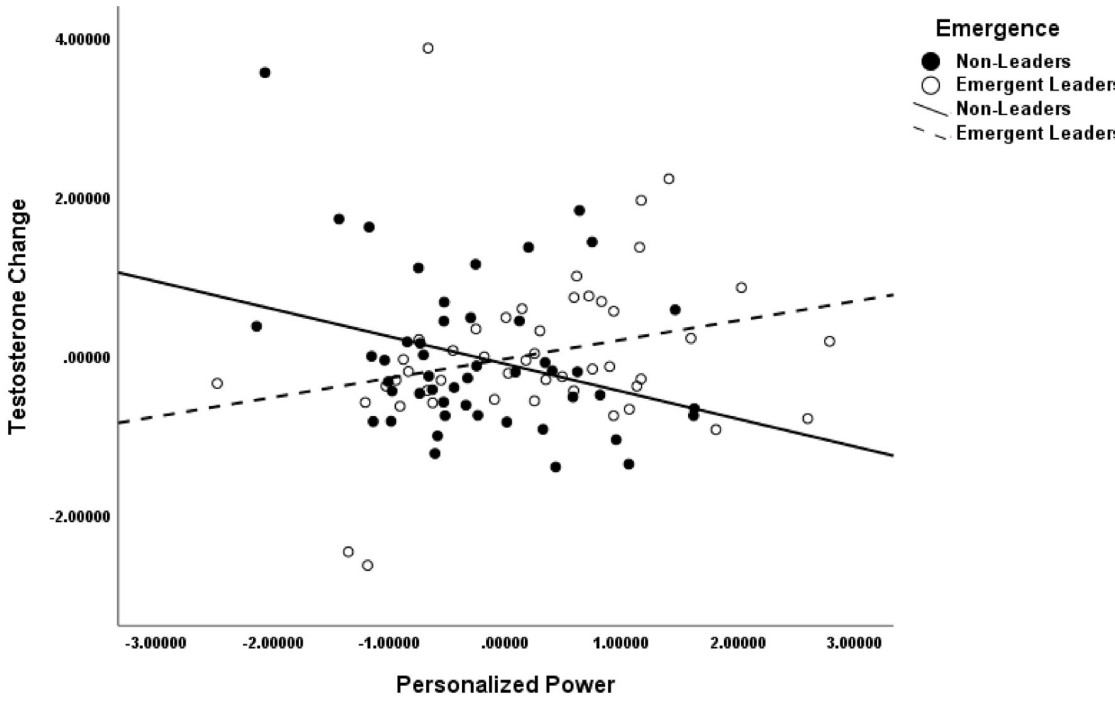

**Fig 2. Interaction of leader emergence and p power on testosterone change.**

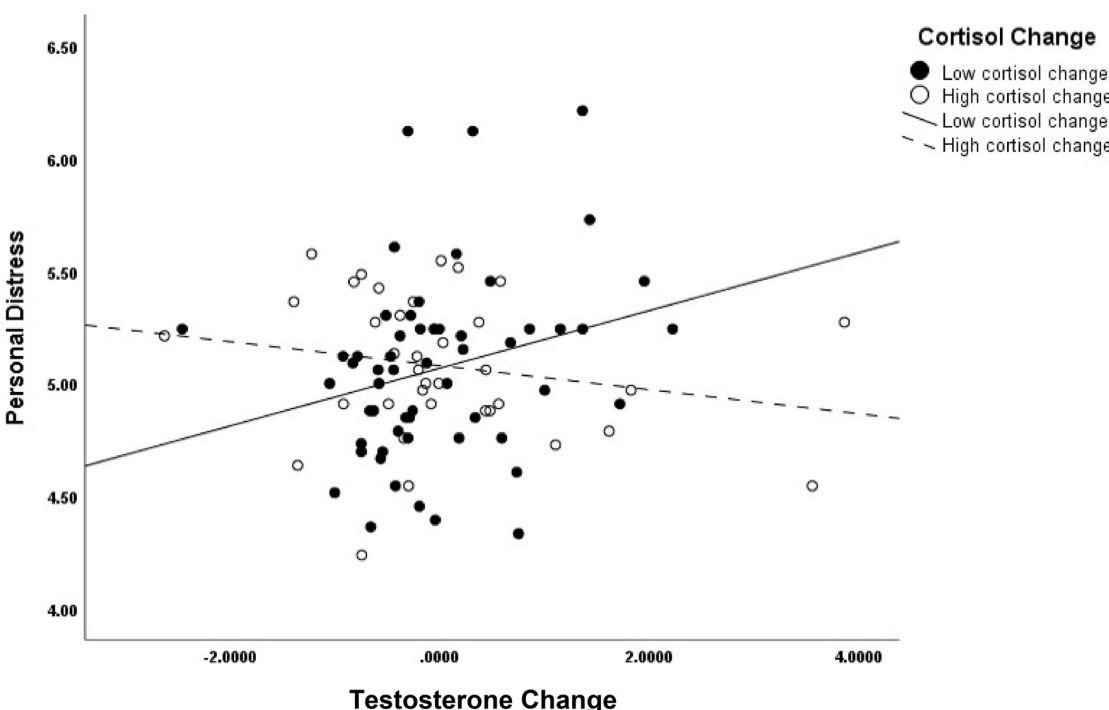

**Fig 3. Interaction of changes in testosterone and cortisol (mean split) on personal distress.**

effect at various moderator levels shows that it is significant at high levels of p Power (+1 *SD*) and again when C change is either low (-1 *SD*) or moderate (between -1 and +1 *SD*).

## Discussion

We examined how leader emergence contributes to changes in personal distress, an important and often understudied facet of empathy [40]. We also tested whether changes in T and C, as well as differences in p Power, could explain how these effects transpire. The results suggest that p Power was a factor in the direction and intensity of T change in emergent leaders, with individuals high in p Power showing a more positive T change compared to those low in p Power.

Scholars noted long ago that there are two types of power motivation each associated with different behavioral manifestations [21, 75, 85, 118, 119]. On the one hand, individuals expressing a personalized power motive (or p Power) are driven by a competitiveness that facilitates the domination of opponents either forcibly or through more subtle means (e.g., persuasion, control). They also display low inhibition or self-control, and have a preoccupation with fame and prestige (e.g., status consumption). Those having a *socialized* power concern (or s Power), however, are more reluctant to admit they enjoy 'having power' over others. Instead, these individuals exercise power through prosocial behaviors intended for others' benefit. Contrary to those high in p Power, these individuals exhibit a more disciplined or inhibited expression of power and, as such, have been shown to be more competent and effective as organizational leaders [for a detailed treatment, see 118]. In designing this study, we focused on arousing p Power instead of s Power because we were more concerned with the processes involved in emergent leadership from a contrived laboratory setting than effective leadership in organizations where followers must be taken into account.

In addition, we investigated whether the dual-hormone hypothesis, which has been predominantly tested using baseline hormone levels [16, 84], would be supported with dynamic testing. If changes in affective outcomes such as personal distress following changes in one's status are attributed to endocrinological reactions, then using hormonal fluxes rather than baselines should provide a more valid test of these mechanisms. This was the case here. That is, no evidence was found for the interaction of baseline T and C to predict changes in personal distress following leader emergence. Using baseline T and C, the T × C interaction predicting personal distress was nonsignificant ($b = .00$, $p = .72$). Similarly, using $T_2$ and $C_2$ yielded a similar nonsignificant interaction ($b = .00$, $p = .46$). Thus, no effect was observed when the more proximal T and C measures, collected at $T_2$, were used.

The dynamic test, however, proved fruitful as changes in C predicted how T reactivity contributed to changes in personal distress, despite findings being in the opposite direction to that anticipated. Originally, we surmised that elevated personal distress characterizes individuals who are low in dominance, and since felt distress is a cue to one's focus on the self rather than on others, it would be seen as a liability in newly appointed leaders [50]. Second, because rises and drops in T and C, respectively, are markers of dominance according to the dual-hormone hypothesis, we also rationalized that T increases should lead to less personal distress at low C-level change. However, rises in T led to *more* personal distress at these levels of C change, not less. One explanation for this finding resides with the notion that personal distress is but one component of empathy, amid several others [39]. For example, a widely used conceptualization of empathy frames it as a cognitive attempt to imagine another person's thoughts and feelings without experiencing any affective response [112]. As such, one can score high or low on any one component of empathy without a parallel score on the other. Perhaps the interaction of the two hormonal changes (i.e., T change × C change) could predict other empathic responses like perspective taking (cognitive empathy) or compassion (behavioral empathy) in the direction that we initially suggested. Research is therefore needed to clarify which type of outcome–affective, cognitive, or behavioral–can be evaluated using the dual-hormone hypothesis.

Another explanation for why increases in T led to increased personal distress at low levels of C change involves questioning the assumption that distress is a sign of submissiveness rather than dominance. There are numerous contexts that may warrant or justify emergent leaders to feel distress. One context comprises situations where a leader's anguish derives from challenging environmental demands such as resource inadequacy or competition that threatens the group's survival. Perhaps in such cases, a leader's distress might be seen as a veritable sign of weakness because followers expect their leaders to be decisive and assertive through such turmoil. Unlike other forms of distress, however, personal distress is an empathic response to the suffering of others. When communicated effectively, displays of such distress could be interpreted as demonstrations of honest concern for the welfare of others, all of which may contribute to perceptions of effective and trustworthy leadership [120]. Perhaps an emergent leader's personal distress could be studied through the lens of costly signaling theory which argues that behaviors appearing to be burdensome or 'costly' to individuals are, in fact, "conspicuous displays of resources that serve to reinforce one's status" [121, p. 81]. In this light, therefore, a leader's personal distress may be deemed a strength rather than a weakness. Many theories from relational to transformational leadership emphasize the value of empathy [122, 123]. If this is the case, then the dual-hormone hypothesis should predict that personal distress (opposite to other forms of distress signaling weakness) be associated with an increase in T only when T's effect is not blocked by a subsequent boost in C production. At high C change, this association should be reversed and personal distress should decrease. However, the intensity of personal distress must be taken into account, as such a relationship is not likely to occur

when personal distress is excessive. High levels of personal distress should not be associated with sustained perceptions of capable leadership because they are likely to contribute to reductions in cognitive and emotional resources [120], as well as having possible detrimental effects on both the sufferer and the observer [38].

Following from the above, our findings show that hormonal factors play a role in individual reactions following leader emergence. Of interest are implications for how organizations can support leaders who emerge informally in work groups, on an affective level, to manage the pressures of their new role. For example, organizations could provide leaders with a means of reflecting on positive work events to help manage the heightened tendency to react to the emotional distress of others following leader emergence [124]. This will decrease the debilitating effects of experienced anxiety without risking the social benefits associated with genuine empathy. Similarly, organizations could examine how emotion regulation intervention strategies [125] can serve to protect leaders from falling into the "emotional curse" wherein leaders are unable to fulfill their vision out of fear of causing some form of follower suffering [126, p. 250]. Lastly, in an effort to promote wellbeing at work, organizations could train newly appointed leaders in effective and healthy emotional expression and management.

## Limitations and future research

The current study is not without its limitations. The focus of our investigation was on emergent male leaders, limiting the generalizability of findings. Researchers have recently argued that T may not serve the same function in both males and females, and that this differentially produces neural and behavioral responses to competitive cues [127]. For example, estradiol plays a significant role in female fertility and contributes to increased dominance motives and status-seeking behaviors in women [128, 129]. Therefore, replacing T with estradiol in our studies might be an avenue worth pursuing when sampling women. More practically, as workplace demographics change, future research would benefit from a more nuanced look at women in leadership positions and the hormonal, psychological, and behavioral processes through which they emerge as leaders [24, 127, 130]. This suggestion concurs with recent critiques on using gender as a simple covariate particularly in hormonal studies [106, 131]. Additionally, future research is encouraged to look beyond T × C interactions to incorporate other physiological variables noted for their involvement in similar socioendocrinological phenomena. For example, Liening and Josephs [132] caution that the neuropeptide hormone arginine vasopressin (AVP) is both dependent on androgens for its synthesis and acts to facilitate the effects of T on instrumental behavior. Specifically, they claim that AVP promotes T-induced behavior by affecting an individual's social perception and preparing them for defensive aggression. However, AVP, has also been known to stimulate the secretion of C [133] which, in turn, dampens T release [90]. Although this triadic hormonal cascade does not threaten the validity of the current findings, future researchers should consider the effects of AVP at various levels when discussing the dual-hormone hypothesis. Such an approach is expected to provide us with an improved understanding of the complex processes that underlie the association between hormones and behavior.

A second limitation involves a potential randomization bias in both studies. We employed an experimental design and an analytic procedure that utilizes bootstrapping to ameliorate some of the shortcomings of other designs regarding the consistency in estimates and inferences [134, 135]. As discussed, emergent leaders scored higher in p Power than their nonleader counterparts, even though the assignment procedure was randomized and time-separated. After statistically controlling for this occurrence, the results were robust and consistent in size, direction, and significance, suggesting that this might have been an artifact of chance. Future

research can use the same pairing procedure for p Power as was used for the NTT competition in which participant scores are matched based on p Power scores with one participant being randomly assigned to the emergent leader condition.

A third limitation rests on the difficulty of teasing apart the observed effects due to the formal process of emerging as a leader or to some other factor characterizing the study (e.g., being praised). As many of these processes are intertwined [136], it becomes difficult to ascertain whether individual effects are at play beyond the use of a manipulation check that asks participants to comment on the extent to which they felt like a leader. Future work in this area could attempt to parcel out these factors by manipulating them in a controlled laboratory setting and by using more formal assessments of leader emergence, all of which would shed more light on causality [135, 137].

Lastly, when scrutinizing the overall hormonal mechanism, we used a measurement-of-mediation rather than a manipulation-of-mediator design [134, 135, 138, 139]. Hence, causal inferences remain stronger in the first half of our model (i.e., leader emergence to T change) due to randomization of the leader emergence condition compared to the second half of the model (i.e., T change to personal distress). Researchers should nevertheless be aware of the complications surrounding hormonal data collection outside a controlled environment, including sample size restriction that inhibits proper subgroup analysis [66] and contextual realities that prohibit the interruption of work and/or workplace interactions. A solution could be to use mediational manipulations in which the hormonal vacillations within leader emergence conditions are manipulated and randomized. For instance, emergent leaders could be randomized into two groups with one receiving exogenous or artificially administered T while the other does not (see the placebo controlled, double-blind crossover designs in [80]).

## Conclusion

To recapitulate, our goal was to investigate the motivational and hormonal underpinnings of the relationship between leader emergence and personal distress. Three developments in psychology over the last half century, in chronological order, shaped our conceptual thinking. The first was the pioneering work of Winter [85] and McClelland [21] on the centrality of implicit power motivation in leadership. The second was the association between social status and T [71, 72], and the third was the comparatively recent research in behavioral endocrinology on T × C interactions [13]. By combining these three areas, we hope to have contributed to the growing body of work on leadership, individual differences, and emotions by shedding light on how leadership attainment, itself a status enhancer in groups, is associated with one form of empathy. Above all, we hope that this research will help encourage future interdisciplinary interest and inquiry.

## Acknowledgments

We thank Pauline Aldon and Rime Iman for their invaluable assistance with logistics during data collection. We are also grateful to Concordia University's Centre for Multidisciplinary Behavioral Business Research and the Laboratory for Sensory Research for their resource support.

## Author Contributions

**Conceptualization:** John G. Vongas.

**Data curation:** John G. Vongas, Raghid Al Hajj.

**Formal analysis:** Raghid Al Hajj.

**Methodology:** John G. Vongas.

**Project administration:** John G. Vongas.

**Resources:** John G. Vongas.

**Software:** John G. Vongas.

**Writing – original draft:** John G. Vongas, Raghid Al Hajj, John Fiset.

**Writing – review & editing:** John G. Vongas, Raghid Al Hajj, John Fiset.

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
