## [Decision Letter · Decision Letter 0]

2 Dec 2020

PONE-D-20-32353

Leader emergence and affective empathy: A dynamic test of the dual-hormone hypothesis

PLOS ONE

Dear Dr. Vongas,

Thank you for submitting your manuscript to PLOS ONE. After careful consideration, we feel that it has merit but does not fully meet PLOS ONE’s publication criteria as it currently stands. Therefore, we invite you to submit a revised version of the manuscript that addresses the points raised during the review process.

Note that we were only able to secure one review, probably due to the difficult times everyone is going through. The comments of this review are attached to this letter. Based on our own reading of the paper, and in the attempt to provide a timely answer, we felt it save to proceed, although we usually try to secure at least two reviews.

We look forward to receiving your revised manuscript.

Kind regards,

Claus Lamm

Academic Editor

PLOS ONE

Journal Requirements:

Reviewers' comments:

Reviewer's Responses to Questions

**Comments to the Author**

1. Is the manuscript technically sound, and do the data support the conclusions?

Reviewer #1: Yes

2. Has the statistical analysis been performed appropriately and rigorously? 

Reviewer #1: Yes

3. Have the authors made all data underlying the findings in their manuscript fully available?

Reviewer #1: Yes

4. Is the manuscript presented in an intelligible fashion and written in standard English?

Reviewer #1: Yes

5. Review Comments to the Author

Reviewer #1: My applause to the authors of the PONE-D-20-32353 mss! The model tested is amazingly relevant in these trying times in which we witness leadership of all forms tested under high stress.

1. The design is clear and relevant both in the research world and in practice.

2. The measures chosen are quite appropriate and their processing was handled well.

3. The data analysis was meticulous and explained well.

4. The results offer another important step in understanding the psycho-physiology of leadership.

5. The choice of personalized power as the key motivational variable is fine for this arousal (experimental) design. Being a part of the original research, it was socialized power that was thought and found to be key to effective leadership. But in this design, the arousal was a perceived zero-sum competition. In such a situation, personalized power would be both the fastest to be aroused and most immediately relevant. A comment about that should be made in the Discussion. That would clarify a possible confusion in thinking about emergent leadership in highly competitive settings versus effective leadership.

6. The endocrinological hypotheses are sound. But there is something more subtle possibly involved. Testosterone has been linked primarily to instrumental behavior. Often, through its constituent role in creating vasopressin, the instrumental causality becomes clear. At modest doses, vasopressin is also linked to Parasympathetic Nervous System arousal which ameliorates the negative effects of stress. Since cortisol is a clear endocrine marker of stress, the interactions found in this study make sense. But at higher doses, both testosterone and vasopressin can drive more, in McClelland's terms, "power stress" with its effect of increasing stress hormones, like cortisol. This is no way threatens the validity or importance of the findings in this study, but it suggests a dosage effect worthy of comment in the Discussion to avoid simplistic conclusions about T and C and leadership effectiveness and sustainability.

6. PLOS authors have the option to publish the peer review history of their article (what does this mean?). If published, this will include your full peer review and any attached files.

Reviewer #1: **Yes: **Richard E. Boyatzis

---

## [Author Response · Author response to Decision Letter 0]

9 Dec 2020

Dear Drs. Lamm and Boyatzis:

Kindly note that these responses are also included in a separate document titled "Responses to Reviewers," which we were asked to upload. For your convenience, we include them here as well. Please allow me to use RB (Reviewer initials) and JV (my initials) as follows:

RB: My applause to the authors of the PONE-D-20-32353! The model tested is amazingly relevant in these trying times in which we witness leadership of all forms tested under high stress.

JV: Thank you for taking the time to review our work. We very much appreciated your insightful comments and your praise.

RB: The design is clear and relevant both in the research world and in practice. 

JV: Thank you.

RB: The measures chosen are quite appropriate and their processing was handled well.

JV: Thank you. The PSE coding was a laborious task that required much attention to detail.

RB: The data analysis was meticulous and explained well.

JV: Thank you.

RB: The results offer another important step in understanding the psycho-physiology of leadership.

JV: Thank you.

RB: The choice of personalized power as the key motivational variable is fine for this arousal (experimental) design. Being a part of the original research, it was socialized power that was thought and found to be key to effective leadership. But in this design, the arousal was a perceived zero-sum competition. In such a situation, personalized power would be both the fastest to be aroused and most immediately relevant. A comment about that should be made in the Discussion. That would clarify a possible confusion in thinking about emergent leadership in highly competitive settings versus effective leadership.

JV: We value this comment and acknowledge that it is important to address this distinction between emergent leadership, where p Power may be particularly useful, and effective leadership in organizations where s Power is key, as your pioneering work has long established. This will clarify any potential confusion on our choice for using one ‘face of power’ of the other. We include this new section in our revised manuscript on page 30 (i.e., the entire second paragraph immediately following the “Discussion” heading).

RB: The endocrinological hypotheses are sound. But there is something more subtle possibly involved. Testosterone has been linked primarily to instrumental behavior. Often, through its constituent role in creating vasopressin, the instrumental causality becomes clear. At modest doses, vasopressin is also linked to Parasympathetic Nervous System arousal which ameliorates the negative effects of stress. Since cortisol is a clear endocrine marker of stress, the interactions found in this study make sense. But at higher doses, both testosterone and vasopressin can drive more, in McClelland’s terms, “power stress” with its effect of increasing stress hormones, like cortisol. This is no way threatens the validity or importance of the findings in this study, but it suggests a dosage effect worthy of comment in the Discussion to avoid simplistic conclusions about T and C and leadership effectiveness and sustainability.

JV: This was also a notable comment and we agree that we should have better articulated how complex these hormonal mechanisms truly are. In the revised manuscript, we allude to vasopressin (arginine vasopressin or AVP) and its effect on both T and C. This section is found on page 34, added at the end of the first paragraph following the heading “Limitations and future research.” We are also aware of McClelland’s research on the relationship between AVP and the need for achievement (nAch) but opted to leave this detail out of the manuscript because our study does not address implicit motives other than power.

---

## [Editor Report · Decision Letter 1]

14 Dec 2020

Leader emergence and affective empathy: A dynamic test of the dual-hormone hypothesis

PONE-D-20-32353R1

Dear Dr. Vongas,

We’re pleased to inform you that your manuscript has been judged scientifically suitable for publication and will be formally accepted for publication once it meets all outstanding technical requirements.

Kind regards,

Claus Lamm

Academic Editor

PLOS ONE
---

## [Editor Report · Acceptance letter]

18 Dec 2020

PONE-D-20-32353R1 

Leader emergence and affective empathy: A dynamic test of the dual-hormone hypothesis 

Dear Dr. Vongas:

I'm pleased to inform you that your manuscript has been deemed suitable for publication in PLOS ONE. Congratulations! Your manuscript is now with our production department. 

Kind regards, 

on behalf of

Dr. Claus Lamm 

Academic Editor

PLOS ONE